# Time Dependent Changes in the Ovine Neurovascular Unit; A Potential Neuroprotective Role of Annexin A1 in Neonatal Hypoxic-Ischemic Encephalopathy

**DOI:** 10.3390/ijms24065929

**Published:** 2023-03-21

**Authors:** Hyun Young Park, Valéry L. E. van Bruggen, Carine J. Peutz-Kootstra, Daan R. M. G. Ophelders, Reint K. Jellema, Chris P. M. Reutelingsperger, Bart P. F. Rutten, Tim G. A. M. Wolfs

**Affiliations:** 1Department of Pediatrics, School of Oncology and Reproduction (GROW), Maastricht University, 6229 ER Maastricht, The Netherlands; 2Department of Psychiatry and Neuropsychology, School for Mental Health and Neuroscience (MHeNs), Maastricht University, 6229 ER Maastricht, The Netherlands; 3Department of Pathology, Gelre ziekenhuizen, 7334 DZ Apeldoorn, The Netherlands; 4Department of Pediatrics, Maastricht University Medical Centre, 6229 ER Maastricht, The Netherlands; 5Department of Biochemistry, Cardiovascular Research Institute Maastricht (CARIM), Maastricht University Medical Center, 6200 MD Maastricht, The Netherlands

**Keywords:** hypoxic-ischemic encephalopathy, blood-brain barrier (BBB) integrity, neurovascular unit (NVU), Annexin A1, therapeutic window

## Abstract

Perinatal brain injury following hypoxia-ischemia (HI) is characterized by high mortality rates and long-term disabilities. Previously, we demonstrated that depletion of Annexin A1, an essential mediator in BBB integrity, was associated with a temporal loss of blood-brain barrier (BBB) integrity after HI. Since the molecular and cellular mechanisms mediating the impact of HI are not fully scrutinized, we aimed to gain mechanistic insight into the dynamics of essential BBB structures following global HI in relation to ANXA1 expression. Global HI was induced in instrumented preterm ovine fetuses by transient umbilical cord occlusion (UCO) or sham occlusion (control). BBB structures were assessed at 1, 3, or 7 days post-UCO by immunohistochemical analyses of ANXA1, laminin, collagen type IV, and PDGFRβ for pericytes. Our study revealed that within 24 h after HI, cerebrovascular ANXA1 was depleted, which was followed by depletion of laminin and collagen type IV 3 days after HI. Seven days post-HI, increased pericyte coverage, laminin and collagen type IV expression were detected, indicating vascular remodeling. Our data demonstrate novel mechanistic insights into the loss of BBB integrity after HI, and effective strategies to restore BBB integrity should potentially be applied within 48 h after HI. ANXA1 has great therapeutic potential to target HI-driven brain injury.

## 1. Introduction

Neonatal brain injury following hypoxia-ischemia (HI), known as Hypoxic-Ischemic Encephalopathy (HIE), is characterized by high mortality rates and lifelong morbidities, such as impaired motor and/or neurocognitive function [1,2,3,4]. The molecular and cellular mechanisms mediating the impact of HI are not fully scrutinized.

Loss of integrity of the blood-brain barrier (BBB) is one of the important factors that contribute to perinatal brain injury in HIE [5]. The protective function of the BBB is carried out by the neurovascular unit (NVU), which consists of the basement membrane (BM) and a variety of cells, such as endothelial cells, pericytes, and astrocytes [6]. Structural proteins of the BM, such as laminin and collagen type IV, facilitate essential communication between all NVU components via cell-matrix interactions, as illustrated in Figure 1 [7]. This close collaboration between cells and matrix components of the NVU is essential for architectural support, the supply of molecules, signal transduction and maintenance of cerebral homeostasis [8,9,10]. Therefore, loss of the barrier function of the BBB could disrupt cerebral homeostasis and expose the brain to injurious mediators.

A hypoxic-ischemic insult can increase BBB permeability in the initial period of HI, followed by a reperfusion phase where the subsequent release of reactive oxygen species and pro-inflammatory cytokines can provoke additional injury [11,12]. In a previous manuscript, we demonstrated in these animals that a global HI insult compromised the BBB integrity with albumin leakage into brain parenchyma after 72 h [13], followed by infiltration of peripheral immune cells with a subsequent enhanced cerebral inflammatory response 7 days after global HI [14]. At present, the underlying mechanisms for this loss of BBB function following global HI remain largely unclear.

Recently, we identified the immunomodulatory protein Annexin A1 (ANXA1) as a promising therapeutic candidate for HIE. ANXA1 is an essential endogenous regulator of BBB integrity in neurodegenerative diseases because it strengthens endothelial tight junction formation through the formyl peptide receptor 2 (FPR2)/RhoA pathway. In addition, there is compelling evidence for the role of ANXA1 in several anti-inflammatory processes, including neutrophil migration, macrophage phagocytosis, shifting microglial phenotype from M1 to M2, and releasing pro- or anti-inflammatory mediators [15,16,17]. We found a rapid and temporary decrease in endogenous ANXA1 expression in the cerebrovasculature preceding a loss of BBB integrity after HI in the ovine fetus [13], which is considered to be in part the mechanistic explanation behind the observed functional BBB loss following HI. Interestingly, this concept is confirmed and extended in vivo in ANXA1-/- mice, in which the absence of ANXA1 affected BBB integrity through disruption of tight junctions and changes in actin cytoskeleton [16]. However, the effect of ANXA1 depletion on basement membrane composition and other cells of the NVU remains unknown.

Preventing loss of BBB integrity at an early stage after global HI could potentially protect the brain from additional inflammatory injury in HIE [3,18,19]. Therefore, increasing knowledge of the structural changes of the BBB after HI can lead to the identification of potential therapeutic targets such as ANXA1. The aim of this study was to gain further insight into the composition of essential structural and cellular components of the NVU following global HI in a translational HIE model. We hypothesize that ANXA1 depletion after global HI in fetal sheep is associated with subsequent structural and cellular changes in the NVU. We hypothesize that the global HI driven loss of ANXA1 expression negatively affects essential ovine BBB structures and supporting cells of the NVU. 

## 2. Results

### 2.1. Global Hypoxia-Ischemia Is Associated with a Significant Depletion of ANXA1 Expression One Day after HI, Leading to Laminin Depletion Three Days Post-HI at the Neurovascular Unit (NVU) in the Ovine Fetus

We first investigated the effect of global HI on ANXA1 expression in the cerebrovasculature. To assess the potential time-dependent alterations, analyses were performed 1, 3, and 7 days after HI. Coronal sections of the preterm ovine brain were analyzed for ANXA1 immunoreactivity (IR) as illustrated in Figure 2. One day after global HI, endogenous ANXA1 IR was significantly reduced in blood vessels compared to controls (1d sham vs. 1d HI, *p* = 0.032) (Figure 2a,b), confirming our previous data [13].

Laminins are a major constituent of the basement membrane and an excellent marker to study cerebrovascular integrity [20,21]. Importantly, we found that a loss of ANXA1 at 1 day post HI was followed by a decrease in laminin expression at 3 days compared with time-matched control groups (3d sham vs. 3d HI; *p* = 0.057), confirmed by integrated density analysis (3d sham vs. 3d HI; *p* = 0.057). Additionally, upon 7 days of global HI, laminin IR was increased in comparison with 1 and 3 days post HI in both scoring analyses (1d HI vs. 7d HI; *p* = 0.043, 3d HI vs. 7d HI; *p* = 0.009) and integrated density between 3d and 7d global HI (3d HI vs. 7d HI; *p* = 0.017) (Figure 2a–d).

### 2.2. Global HI Induced an Increased Microvascular Collagen Type IV Expression at the NVU Seven Days Post-Reperfusion in the Ovine Fetus

We investigated type IV collagen, a molecular component of the basal lamina, which creates structural stability between multiple BM components [22,23,24,25]. The type IV collagen score and integrated density were increased at 7 days post HI compared with time-matched controls (7d sham vs. 7d HI; *p* = 0.019, *p* = 0.057) (Figure 3a,b). Both scoring and integrated density showed upregulation in type IV collagen comparing 3 and 7 days post HI, with the score analysis reaching statistical significance at 3d HI vs. 7d HI; *p* = 0.024, *p* = 0.094) (Figure 3a). 

### 2.3. Global HI Induced Upregulation of Pericyte Coverage in the NVU Seven Days Post-Reperfusion in the Ovine Fetus

Pericytes are vascular mural cells that are in direct contact with endothelial cells in the microvasculature of the CNS, supporting the barrier function of the BM [26]. We used platelet-derived growth factor receptor beta (PDGFRβ) as an immunohistochemical marker to study the pericyte coverage of the BBB [27,28,29]. We found that pericyte coverage was upregulated at 7 days post HI compared with time-matched controls (7d sham vs. 7d HI; *p* = 0.056), but not after 1 or 3 days (1d sham vs. 1d HI; *p* = 0.222, 3d sham vs. 3d HI; *p* = 0.99). Additionally, we assessed the time-dependent pericyte coverage after an HI insult. Pericyte coverage increased significantly 7 days post HI compared with 3 days post HI (3d HI vs. 7d HI; *p* = 0.05) (Figure 4a).

## 3. Discussion

Perinatal brain injury after hypoxic-ischemia (HI) is a major health concern as newborns can develop long-term neurologic sequelae, with blood-brain barrier (BBB) dysfunction as an important contributor [1,3]. There is limited understanding of HI driven alterations of the neurovascular unit (NVU) as part of the BBB in neonates, which we aim to address in this study.

We have demonstrated in the same animal model that global HI leads to a transient depletion of ANXA1, followed by loss of BBB integrity, demonstrated by albumin extravasation in the brain parenchyma [13,30]. ANXA1 is suggested to strengthen the BBB integrity by FPR2-dependent inhibition of RhoA kinase and cytoskeleton stabilization, leading to tight junction stabilization between endothelial cells [15,16]. The importance of ANXA1 in BBB integrity is supported by its neuroprotective capacities in the periphery of the brain [16] and in neurological diseases, including Alzheimer’s disease (AD). Recently, Ries et al. showed that administration of ANXA1 reversed the significant decrease in laminin in a mouse model for Alzheimer’s disease [31]. Moreover, in a murine metabolic syndrome model, a compromised BBB was detected, including disruption of tight junctions and a decrease in laminin expression. The administration of ANXA1 repaired BBB integrity loss by increasing laminin expression, even during maintenance of the detrimental high-fat, high-sugar diet [32]. Here, we confirmed and extended these findings by demonstrating that depletion of ANXA1 within 24 h after hypoxia-ischemia in fetal lambs was associated with disruption of laminin expression 3 days after HI induction. In line with this, other studies have found laminin to be affected in adult rodent models of HI and cerebral stroke [33,34]. These combined findings underline the importance of ANXA1 as a neuroprotective agent in maintaining BBB integrity in the context of neonatal hypoxic-ischemia and ANXA1 driven stabilization of laminin expression in the neurovascular unit as an essential mechanism herein. Importantly, although we did identify these time dependent changes of ANXA1 and laminin, it remains at present unclear whether (1) ANXA1 levels reach the lowest point at the studied timepoint (being 24 h post UCO), (2) when the replenishment of ANXA1 is initiated between 24–72 h post UCO, and (3) whether the observed decrease in laminin is initiated before the studied timepoint (3 days post UCO). In other words, we found an association between a temporary loss of ANXA1 and a decrease in laminin, which is supported by the literature, but the decrease in ANXA1 and the subsequent drop of laminin might follow one after another faster than can be derived from the current data, where we only studied expression levels at three timepoints. This assumption is supported by earlier findings in other hypoxic-ischemic studies, which demonstrated ANXA1 and laminin to be downregulated up to two days post-HI [35,36,37]. Within this context, our data suggest the involvement of a potential recovery mechanism behind replenishment of ANXA1 levels after a hypoxic-ischemic (HI) insult. First, it is tempting to speculate that glucocorticoids are involved in correcting ANXA1 levels. More precisely, upon a HI insult, the hypothalamic pituitary adrenal (HPA) axis stimulates the adrenal glands to secrete glucocorticoids within the first 24 h [38], and glucocorticoids have been well described as regulators of ANXA1 [39,40,41,42]. Alternatively, other mechanisms might be responsible for regulating ANXA1 levels, including miRNAs [43,44]. Both a more detailed dynamics of ANXA1 and laminin including the potential involvement of regulators of ANXA1 expression will be addressed in a follow up study.

The observed changes in laminin composition in our model are paralleled by alterations in collagen type IV expression seven days after HI induction. While laminin plays a key role in interacting with cell-surface proteins, type IV collagen creates a distinct matrix by interacting with laminin, together maintaining the stability of the BM [45,46,47]. The synchronized expression pattern of laminin and collagen IV was previously identified in vitro, in a model of hypoxia-induced BBB dysfunction [48]. The initial decrease in laminin is followed by an increase, concomitant with increased collagen type IV expression, suggesting that significant remodeling occurs in the NVU in the first seven days after HI. The essential role of extra-cellular matrix proteins such as laminin and collagen IV in remodeling has been previously described in the context of cerebral stroke and neurodegenerative diseases [49,50]. Interestingly, the restored expression of laminin and increase in collagen type IV expression are paralleled by changes in pericyte coverage in our model. Pericytes are necessary for many stages of vascular remodeling, affecting brain microvascular integrity [51,52,53]. Furthermore, pericytes produce a specific laminin responsible for NVU integrity [20,54], and our findings suggest that the increased pericyte presence with laminin expression provide the needed stability to the damaged BBB. In our model, temporal changes in pericyte coverage included a significant increase seven days after HI. This increase in pericyte coverage is accompanied by an increase in laminin and collagen type IV expression, suggesting that compensatory vascular remodeling of the neurovascular unit is initiated from 96 h post HI onwards. In support of this theory, the observed time related changes of pericytes, are in line with previous studies on adult stroke in rodent models [55], reviewed by Di et al. [56]. In addition to the structures supporting the cerebral microvasculature, which were investigated in this study, endothelial cells are also important in BBB integrity and remodeling processes [57]. More precisely, there is bidirectional crosstalk between endothelial cells and pericytes. For instance, during HI, pericytes stimulate endothelial cell survival via paracrine VEGF-A signaling [58], whereas endothelial cells secrete PDGF-BB, a ligand for pericyte-receptor PDGFR-β, regulating and redirecting pericyte survival and migration to injured sites [29,59,60]. Moreover, essential basement membrane structures such as collagen type IV and laminin, which are highly regulated by endothelial cells [61], affect barrier function by elevating transcellular resistance in vitro [62], and by regulating the expression of tight junction proteins such as occludin [63]. These examples highlight the intricate organization of the NVU and the role for endothelial cells herein. In upcoming studies, a more detailed and functional assessment of endothelial cells in the context of HI will be addressed.

Studies in adult neuropathology such as neurodegeneration, traumatic brain injury, or seizures show similar time-dependent changes of extracellular matrix expression as we have presented in this study (reviewed in [49]). Although the compensatory adaptation of the NVU prevents further brain injury in the acute phase, remodeling of the neurovascular unit could have detrimental long-term effects on neurovascular functioning, which is especially important in neonates. Many studies have addressed the possible negative long-term effects of remodeling on endothelial function, neurovascular coupling, and neuro-immune crosstalk leading to adverse neurodevelopmental outcomes [64]. In line with this, previous studies in adult cerebral stroke confirm that therapeutic interventions before NVU remodeling are essential for preventing further cerebral damage, which further underpins the importance of timely intervention to counteract the loss of BBB integrity and its negative sequelae, leading to the disease process of HI in the neonatal brain [65,66,67]. Within this context, this study reveals a potential critical therapeutic window of 24–48 h post hypoxia, considering the observed detrimental mechanistic changes of the neurovascular unit within 72 h after HI and initiation of the compensatory remodeling of the NVU after 96 h [13].

In conclusion, in an ovine model for hypoxic ischemia, we demonstrated dynamic loss of BBB integrity and potential subsequent remodeling, which are reported to be associated with decreased neurocognitive outcome. We present a critical therapeutic window where structural alterations are initiated in the NVU after HI. This study further endorses the potential protective role of ANXA1, in particular in promoting neurovascular integrity and BBB stability. We propose that the time interval within 48 h seems optimal for effective strategies to restore BBB integrity in HI-related brain injury during development, of which ANXA1 could have great therapeutic potential.

## 4. Materials and Methods

### 4.1. Experimental Design and Study Approval

This study examined fetal sheep between 100 and 110 days of gestation, which accounts for approximately 70% of the ovine gestation. At this period of gestation, ovine brain development is equivalent to that of the human brain between 28 and 30 weeks of gestation [20]. Single fetuses (*n* = 37) of Texel pregnant ewes were randomly assigned to two experimental groups: (1) Sham UCO (umbilical cord occlusion) treatment and (2) UCO treatment (HI). At 102 days of gestational age (GA), fetuses underwent general anesthesia and received venous and arterial catheters and a vascular occluder was placed around the umbilical cord. Before surgery, ewes received prophylactic antibiotics (1000 mg amoxicillin and 200 mg clavulanic acid). A catheter was inserted into the maternal saphenous vein to provide access for peri-operative saline infusion (250 mL/hour), post-operative blood sampling and the administration of prophylactic antibiotics during recovery. Ewes were anesthetized by induction of 1 to 2% isoflurane guided by depth of sedation in combination with remifentanil i.v. (0.75 μg/kg/min) for analgesia. Certified personnel monitored vital parameters and the depth of sedation. After surgery, sheep were individually housed with free access to food and water *ad libitum*. 

After four days of recovery, transient global hypoxia-ischemia (HI) was induced by occluding the umbilical cord for 25 min by inflation of the vascular occluder (OCD16HD, 16 mm; In Vivo Metric, Healdsburg, CA, http://www.invivometric.com, accessed on 14 June 2021) or sham occlusion. Fetuses were delivered through caesarean section and sacrificed 1 day (*n* = 10), 3 days (*n* = 8) or 7 days (*n* = 13) after (sham) UCO and immediately prepared for tissue sampling (Figure 5). At the end of each experiment, both the ewe and the fetus were euthanized by administering pentobarbital (200 mg/kg). All tissue sampling and post-mortem analyses were conducted in a blinded manner. All experimental procedures were approved by the Animal Experiments Committee of Maastricht University Medical Center, NL (DEC 2012-064) and conducted with ARRIVE guidelines (https://arriveguidelines.org/, accessed on 10 July 2021) and the Maastricht University guidelines on the Care and Use of the Laboratory Animals.

### 4.2. Immunohistochemistry and Analysis

Fetal brains were dissected and weighed immediately after sacrifice. The brain was halved along the midline, and the right cerebral hemisphere was submerged and fixed in cold 4% paraformaldehyde for three months, further processed through alcohol and xylene washes. After fixation, predefined coronal sections were embedded in paraffin and cut into 4 μm thick sections using a Leica RM2235 microtome (Leica Microsystems B.V., Amsterdam, The Netherlands) to perform immunohistochemical studies. Immunohistochemical staining was performed on adjacent slides within the predefined striatal regions containing basal ganglia and the lateral ventricle for each animal. Endogenous peroxidase activity was blocked by incubating tissue slides in 0.3% hydrogen peroxide dissolved in Tris-Buffered Saline (TBS) for 20 min. Antigen retrieval was performed by boiling tissue slides in sodium citrate buffer (pH 6.0) using a microwave oven. The following primary antibodies were administered on tissue slides; polyclonal rabbit ANXA1 (ab137745, Abcam, Cambridge, UK, 1:100), laminin (ab11575, Abcam, 1:100), Collagen IV (ab6586, Abcam; ratio 1:400), and PDGFRβ (#3169, Cell signaling; 1:100) at 4 °C overnight. Thereafter incubation with a secondary antibody followed; polyclonal swine anti-rabbit biotin (Dako, Santa Clara, CA, USA; 1:200) for anti-ANXA1, goat anti-rabbit HRP (111-035-045, Jackson Immunoresearch) for Collagen IV, and polyclonal donkey anti-rabbit biotin (711-065-152, Jackson Immunoresearch, 1:100) for laminin and PDGFRβ. The antibody specific staining for ANXA1, laminin, and PDGFRβ was enhanced with a Vectastain ABC peroxidase elite kit (Vector Laboratories, Burlingame, CA, USA), followed by a 3,3’-diaminobenzidine (DAB) staining. Counterstaining was performed using Mayer’s hematoxylin, followed by dehydration with increasing alcohol concentration and xylol. The images of ANXA1, Laminin, Collagen type IV, and PDGFRβ were taken at a magnification of 400× with a confocal microscope (Leica DMI 4000, Leica Microsystems, Wetzlar, Germany).

Digital scans were analyzed with Leica Qwin Pro v3.5.1. Software (Leica, Rijswijk, The Netherlands, http://www.leicabiosystems.com, accessed on 1 July 2021) and with QuPath (version v0.2.3, University of Edinburgh, Edinburgh, UK). Cerebral blood vessels were selected within regions of interest (ROI) being periventricular white matter for the analyses of ANXA1 staining [13], and all brain regions, including white and grey matter, were considered for the analysis of laminin, collagen type IV, and PDGFRβ staining. The blood vessels to be analyzed were selected based on a number of criteria. Only vessels bigger than 10 μm as well as smaller than 100 μm were included, following the definition of microvasculature [68]. In addition, any non-perpendicularity blood vessels were avoided to exclude intensity differences. A sub-analysis was conducted to determine the optimal number of blood vessel counts. Two independent and blinded researchers completed all measurements (subjective scorings, semi-quantitative integrated density). Since literature suggests that the spatio-temporal expression patterns of biomarkers such as laminin relate to their function, expression patterns were scored based on the intensity and distribution of immunoreactivity (IR) [69]. Scoring was performed for all makers (ANXA1, laminin, collagen type IV, and PDGFRβ) as the following: 1 as minor, 2 as moderate and heterogeneous expression pattern, and 3 as intense IR with homogenous expression surrounding blood vessels (Figure 6). Then, all qualitative scorings were complemented by a semi-quantitative analysis of area fraction, expressed as the percentage of positive staining area compared with the total tissue area using a standard threshold intensity. A higher mean integrated density indicates higher expression levels of ANXA1, laminin, and collagen type IV. 

### 4.3. Statistical Analysis

All values regarding immunohistochemical staining were shown as standard errors of the mean (SEM) with a 95% confidence interval (CI). All statistical analyses were performed with GraphPad Prism (version 9.1.0, GraphPad Software Inc., La Jolla, CA, USA). The datasets were tested for normality and lognormality. Data showing normal distributions were analyzed using an unpaired, two-tailed Student’s t-test and a one-way analysis of variance (ANOVA). A log-transformation was performed in cases of positive skew, or non-parametric tests were considered. At *p* ≤ 0.05, differences were considered statistically significant, and results are reported with * *p* < 0.05, ** *p* < 0.01, *** *p* < 0.001, and # *p* < 0.06 was considered an effect with biological relevance.

## Figures and Tables

**Figure 1 ijms-24-05929-f001:**
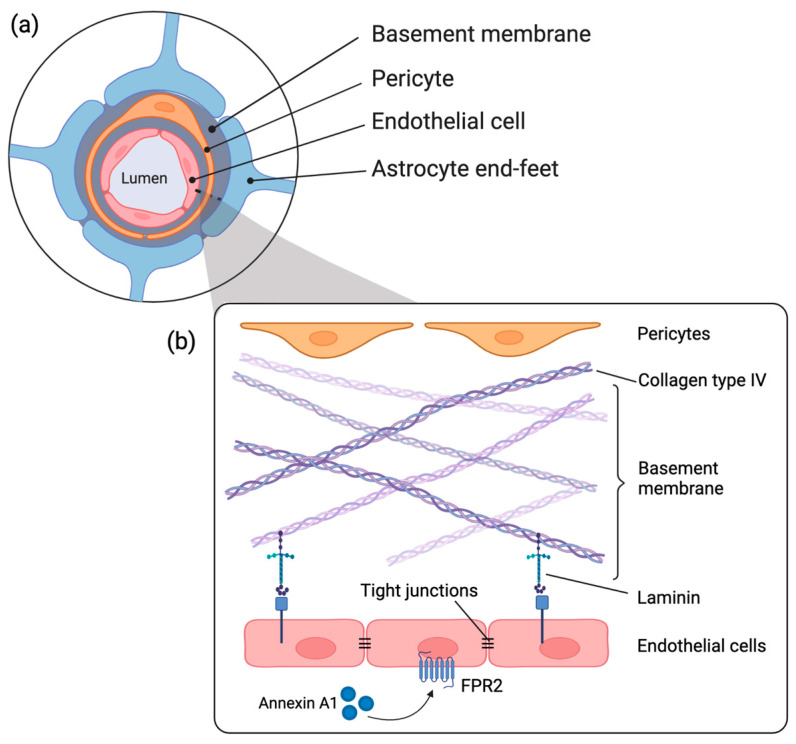
(**a**) Schematic representation of the NVU including vascular cells (endothelial cells, pericytes), basement membrane, and astrocyte end-foot in the CNS; (**b**) structural illustration of the basement membrane components including laminin and collagen type IV, endothelial cells and the pro-resolving signature of Annexin A1 binding to the formyl peptide receptor 2 (FPR2). Created with Biorender.com.

**Figure 2 ijms-24-05929-f002:**
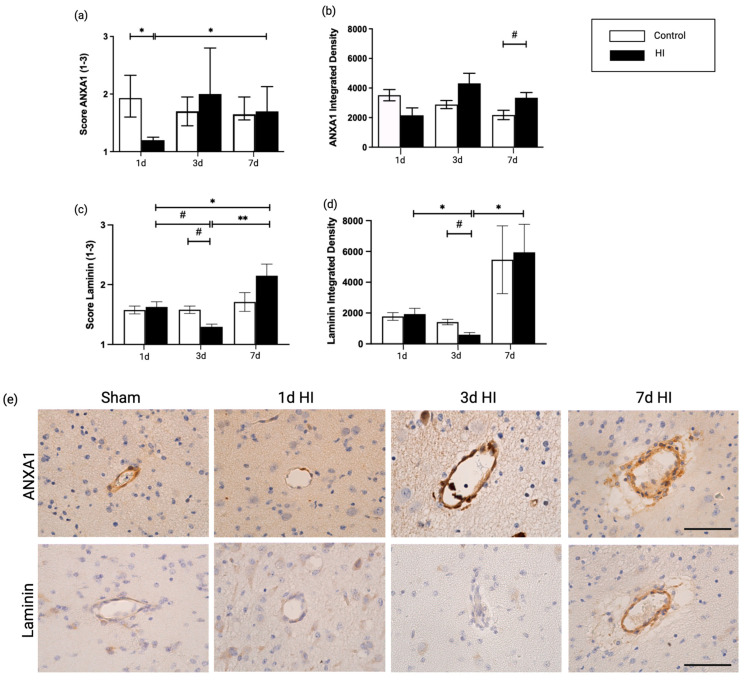
ANXA1 and laminin qualitative scoring (**a**,**c**), and integrated density measurements (**b**,**d**) on 1, 3 and 7 days after HI insult in the cerebral microvasculature; (**e**) representative images of consecutive sections of cerebral blood vessels stained with ANXA1 and laminin. An average of 40 fields of view per animal were analyzed. Graphs show the average integrated density per field of view (mean grey value of stained area × percentage of stained area) for ANXA1 and laminin expression in coronal brain sections. Statistical analysis was performed with a Kruskal-Wallis test followed by Dunn’s post hoc test. Bars represent mean ± SEM. Scale bar 50 μm. 400× magnification * *p* < 0.05, ** *p* < 0.01, # *p* = 0.06.

**Figure 3 ijms-24-05929-f003:**
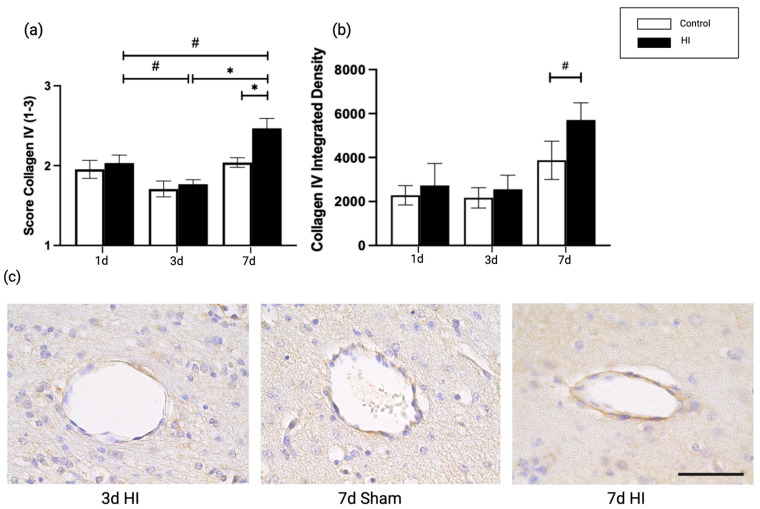
Collagen type IV (**a**) qualitative scoring; (**b**) integrated density measurement on 1, 3 and 7 days after HI; (**c**) representative images of consecutive brain sections of cerebral blood vessels stained with collagen type IV. An average of 40 fields of view per animal were analyzed. Graphs show the average integrated density (mean grey value of stained area × percentage of stained area) of collagen type IV expression in coronal brain sections. Statistical analysis was performed using a Kruskal-Wallis test followed by Dunn’s post hoc test. Bars represent mean ± SEM. Scale bar 50 μm. 400× magnification * *p* < 0.05, # *p* =0.06.

**Figure 4 ijms-24-05929-f004:**
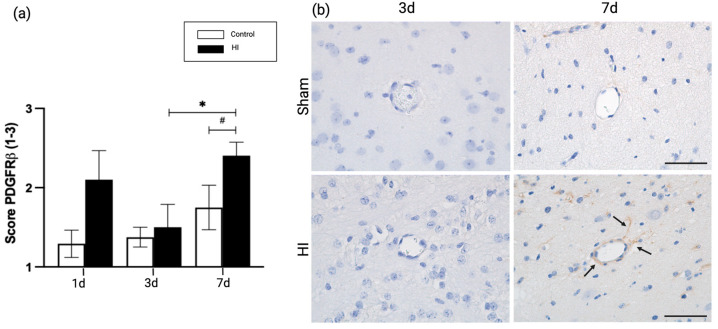
Pericyte coverage surrounding brain microvasculature. (**a**) Qualitative scoring of PDGFRβ at 1, 3 and 7 days post HI or sham; (**b**) Representative microscopic images of PDGFRβ in the microvasculature at 3 and 7 days sham and HI treatment. Arrows indicate pericytes surrounding capillaries at 7d HI. Statistical analysis was performed using a Kruskal-Wallis test followed by Dunn’s post hoc test. Bars represent mean ± SEM. Scale bar 50 μm. 400× magnification * *p* < 0.05, # *p* = 0.06.

**Figure 5 ijms-24-05929-f005:**
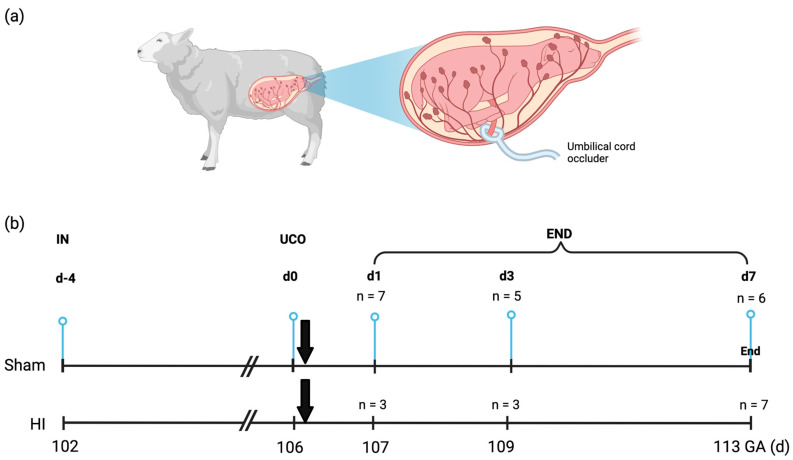
(**a**) Fetal sheep model with placental cotyledons (red dots) and umbilical cord (occluder) depicted; (**b**) Experimental design. Fetuses were instrumented at GA 102 (d-4) followed by 25 min of umbilical cord occlusion (UCO) or sham occlusion (d0) after four days of recovery. On d1, d3 and d7 fetuses were sacrificed and brain tissue was collected. Abbreviations: END—end of experiment; UCO—umbilical cord occlusion; GA—gestational age; HI—Hypoxia-Ischemia; SAL– saline; IN—instrumentation. Created with Biorender.com.

**Figure 6 ijms-24-05929-f006:**
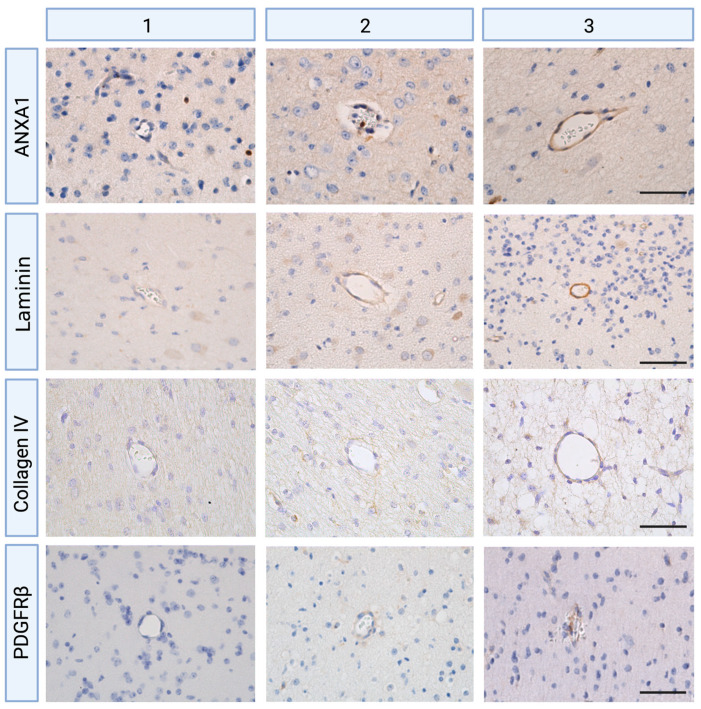
Scoring systems of ANXA1, laminin, collagen type IV, and PDGFRβ in cerebral blood vessels (1 = minor, 2 = moderate, 3 = intense immunoreactivity (IR), magnification 400×, scale bar 50 μm).

## Data Availability

Not applicable.

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
