# Peer review of "Time Dependent Changes in the Ovine Neurovascular Unit; A Potential Neuroprotective Role of Annexin A1 in Neonatal Hypoxic-Ischemic Encephalopathy"

_ijms, 2023, doi:10.3390/ijms24065929_

Round 1
Reviewer 1 Report
This manuscript is titled "Time-dependent changes in the ovine neurovascular unit; a potential neuroprotective role of Annexin A1 in neonatal hypoxic-ischemic encephalopathy" identifies an important role for ANXA1 in the loss of BBB integrity after HI and preliminarily explores the critical time phase for targeting ANXA1 therapy. The present study has important clinical implications, but the relevant conclusions and mechanisms are not rigorous and sufficient, and further refinement of additional experiments is needed.
1. Line 21, ”Result 2.1". I think this result lacks rigor and should be proven in part with ANXA1 knockout mice, or siRNA for animals, and with molecular biology experiments.
2. Line 230-232, In response to this conclusion, the authors should conduct some intervention-targeted therapy experiments to make a preliminary proof.
Reviewer 2 Report
There is an interesting paper. I recommend to describe the methods in details, not to send to previous papers.
Reviewer 3 Report
- In the results section, the authors compared the levels of ANXA1 and Laminin after 1,3 and 7 days after the insult of Hypoxia-ischemia. Based on their results, reported in figure 2, I have several questions regarding these results:
1- Why the levels of ANXA1 are reduced after 1 day post HI and increase after 3 and 7 days (specially after 3 days)? Is there any recovery mechanisms for correcting the levels of ANXA1 and thus reverse the HI?
2- The laminin levels are reduced after 3 days of HI insult. The authors states and I quote: " we found that a loss of ANXA1 at 1 day post HI was followed by a decrease in laminin expression at 3 days...". I can't understand why laminin will decrease after 3 days post HI specially that the levels of ANXA1 are increased? Why the disruption of the integrity of the basement membrane will appear after 3 days of HI specially that ANXA1 is restored and to some extent increase? I would have suspected that this phenomenon would appear more on the 1st day after HI. This concern was discussed in the authors discussion when they stated and I quote: "Ries et al. showed that administration of ANXA1 reversed the significant decrease of laminin in a mice model for AD" which clearly explain that high levels of ANAX1 induce increase levels of Laminin.
3- Regarding collagen type IV and pericytes, I can't understand the increase that appears after 7 days post HI, As for Collagen IV, the levels do not change between the controls and the HI models at 1 and 3 days post HI, which means that collagen IV is not directly inflected by the absence of ANXA1 perturbation. As for pericytes, why they would be activated after 7 days post HI, knowing that the levels of ANXA1 and laminin are restored and therefore the BBB integrity is no more alarming?
Round 2
Reviewer 1 Report
I basically agree with your response and hope to see related research in the future.
Additional questions:
1. Temporal loss of blood-brain barrier (BBB) integrity after HI that has been demonstrated in your animal model? whether the integrity of the BBB is detected in your animal model? Such as Evans-Blue experiment.
2. In FIG1(b), Does it mean blasma membrane of all NVU cells? As we all know that endothelial cells play a major role in maintaining the BBB in NVU. I think that authors should discuss the endothelial cells' function beside pericyte coverage and laminin and collagen type IV, and the effects of these factors on endothelial cells. Although the authors do mention endothelial cells, I think a prominent discussion of the potential functional impact of endothelial cells is needed, or even a focus on endothelial cells in subsequent studies.
Reviewer 3 Report
After reviewing the authors' answers and how they implemented it in the manuscript, I accept to publish the paper in its present form
